

# USING ORDERED WEIGHT AVERAGING (OWA) FOR MULTICRITERIA SOIL FERTILITY EVALUATION BY GIS (CASE STUDY: SOUTHEAST IRAN)

**Marzieh Mokarram[1], Majid Hojati[2]**

[1]Department of Range and Watershed Management, College of Agriculture and Natural Resources of Darab, Shiraz University, Iran, Email: m.mokarram@shirazu.ac.ir

[2]Department of GIS and RS, University of Tehran, Faculty of Geography, Dep. of RS & GIS

**Corresponding author:** Marzieh Mokarram, Tel.: +98-917-8020115; Fax: +987153546476 , Address: Darab,
Shiraz university, Iran, Postal Code: 71946-84471, Email: m.mokarram@shirazu.ac.ir

**Abstract**

The Multi-criteria Decision Analysis (MCDA) and the Geographical Information Systems (GIS) are used to provide more accurate decisions for decision makers in order to evaluate the effective factors of the natural science. One of the popular
algorithms of the multi-criteria analysis is the Ordered Weighted Averaging (OWA). The OWA procedure depends on some parameters which can be specified by means of the fuzzy logic. The aim of this study is to take the advantage of incorporating the fuzzy logic into GIS-based soil fertility analysis by OWA in the west of Shiraz, Fars province, Iran. In fact, different soil fertility maps with
different risk level are prepared in the present study. This study introduces a method for farmers in case of make balance between their budget and their farm soil parameters. A farmer can accept more risk it can use more areas for farming and also the amount of needed budget increases too. For determining the soil fertility maps, the OWA parameters such as potassium (K), phosphor (P), copper (Cu), iron (Fe),
manganese (Mn), organic carbon (OC) and zinc (Zn) were used. After generating the interpolation maps with the Inverse Distance Weighted (IDW), the fuzzy maps were generated by the membership functions for each parameter. Finally, by





utilizing OWA, six fertility maps with different risk levels (degrees of uncertainty) were made. The results show that by decreasing the risk (no trade-off), increasing the risk, more area within the study area was suitable in terms of the soil fertility. Therefore, using OWA can generate many maps with different risk levels. This leads to different managements based on different financial conditions of farmers.

**Key words:** Multicriteria Decision Analysis (MCDA); Ordered weighted averaging (OWA); fuzzy; Soil fertility, west Shiraz, Fars province.

## 1. Introduction

Spatial planning involves decision-making techniques which are associated with the Multi Criteria Decision Analysis (MCDA), the multi-criteria Evaluation (MCE) and other similar techniques. Combining GIS with MCDA methods creates a powerful tool for spatial planning (Malczewski, 1999; Shumilov et al., 2011; Kanokporn & Iamaram, 2011; Belkhiri et al., 2011; Salehi et al., 2012; Feng et al., 2012; Ashrafi et al., 2012). The multi-criteria evaluation may be used to develop and evaluate alternative plans which may facilitate a compromise between interested parties (Malczewski, 1996). In general, the GIS-based soil fertility analysis assumes that a given study area is subdivided into a set of basic units of observation such as polygons or raster. Then, the soil fertility problem involves the evaluation and classification of the areal units according to their fertility for a particular activity. There are two fundamental classes of multi-criteria evaluation methods in GIS: the Boolean overlay operations (non-compensatory combination rules) and the weighted linear combination (WLC) methods (compensatory combination rules). These approaches can be generalized within the framework of the ordered weighted averaging (OWA) (Asproth et al., 1999; Jiang and Eastman, 2000; Makropoulos et al., 2003; Malczewski et al., 2003; Malczewski & Rinner, 2005; Malczewski .,2006). OWA is a family of multi-criteria combination

procedures (Yager, 1988). Conventional OWA can utilize the qualitative statements in the form of fuzzy quantifiers (Yager, 1988, 1996). The main goal of this paper is to produce the land suitability maps according to OWA operators for GIS-based multi-criteria evaluation procedures.

OWA has been developed as a popularization of multi-criteria combination by
Yager (1988). The OWA concept has been extended to the GIS applications by Eastman (1997) as a part of the decision support module in GIS-IDRISI. Subsequently, Jiang and Eastman (2000) demonstrate the utility of the GIS-OWA for land use/suitability problems. The implementation of the OWA concept in IDRISI15.01 resulted in several applications of OWA to environmental and
urban planning problems (Asprothet al., 1999; Mendes & Motizuki, 2001).

Mokarram and Aminzadeh (2010) used OWA for land suitability in Shavur plain, Iran. The results showed that OWA is a multi-criteria evaluation procedure (or the combination operator).

Liu and Malczewski (2013) used the GIS-Based Local Ordered Weighted
Averaging in London, Ontario. In the study area, the aim was to implement the local form of OWA. The local model was based on the range sensitivity principle. The results showed that there were substantial differences between the spatial patterns generated by the global and local OWA methods.

Accordingly, the study area is one of the most important centers of agriculture in
Iran, and the aim of the present study is to prepare the soil fertility maps based on the OWA operators of GIS-based multi-criteria evaluation procedures in the southeast, Iran. In this study, we expect that the selected OWA method is the best method for determining the multi-criteria soil fertility. Concerning the



OWA method, the amount of the soil fertility with different risk levels can be

determined. These determinations are useful for farmers who have different

financial conditions.

**2. Study Area**

This study was carried out in the west of Shiraz, Fars province, Iran. It is an area of

about 100.02 km$^2$, and it is located at latitude of N 29° 31´- 29° 38´and longitude

of E 52° 49´ to 52° 57´ (Figure 1). The altitude of the study area ranges from the

lowest of 1,571 m to the highest of 2,203 m. The main agricultural products are

grain, fruit, and vegetables, whereas the partly wooded mountains are used for 120

pastures. This area has a moderate climate and it has been a regional trade center

for over a thousand years. Shiraz's climate has distinct seasons. This city climate is

categorized as a hot semi-arid one though it is a sort of a hot-summer

Mediterranean climate (Csa). Summers are hot with a July average high degree of

38.8 °C (101.8 °F). Winters are cool with an average of freezing temperature in

December and January. Around 300 mm (12 in) of rain falls each year is measured

almost entirely in the winter months though in some cases the same quantity has

fallen in a single month (as in January 1965 and December 2004). In 2011, Shiraz

population was reported as 2,353,696. The majority of Shiraz populations are

Persian.

Geomorphology of the study area is affected by the physical specification of

different geological formations. Besides, because of its location in the Zagros

Mountains, this region is impressed by geological structures and related factors.

The constructing rocks of this region are defined in 2 parts:

    1- Rocks older than Quaternary which are hard and to some extent compacted.

    2- The quaternary and recent sediments which are loose and can form the

       surface alluvium.



3- Quaternary and Recent Sediments are mainly found in plains among mountains, coastal flats, and so on. These two mentioned zones are somehow similar to each other; however, higher mountains peaks of Zagros hold resistant carbonate rocks, high cliffs, crags and high crests with more than 2500m difference in elevation.

4- Fars area stretches westerly to the border of Kazeroon Fault, easterly to the margin of the imaginary line which separates Bandar-Abbas Hinterland from Fars province, northerly to thrust belt, and southerly to the Persian Gulf coastline. Anticlines of this area have different orientations in the northwest-southeast directions, as well as the east west and the northeast-southwest

orientations (Motiei, 1993).

   5- According to the isobaric contours and potentiometric maps, there is a general hydrodynamic flow from the Zagros Mountains to the Persian Gulf. This hydrodynamic flow varies with topography, anticline geometry, faults and fracture intensity, porosity and permeability. Noticeably, the iso-line

contours follow the hydrodynamic flow (Motiei, 1995).



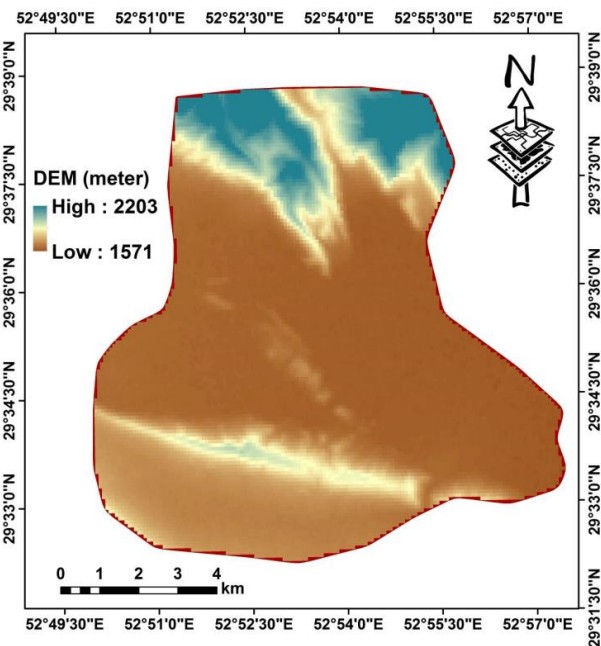

Figure 1. Location of the study area (digital elevation model (DEM) with the spatial resolution of 30 m) (Source: http://earthexplorer.usgs.gov).

The assessment of the soil fertility for the agricultural production process in the region is vital. In running this assessment, environmental factors and human conditions (Soufi, 2004) must be considered. In order to predict the variability of the soil fertility, some minerals were used which are named here as potassium (K), phosphor (P), copper (Cu), iron (Fe), manganese (Mn), organic carbon (OC) and zinc (Zn); then, maps of each parameter were prepared (Table 2) (Organization of Agriculture, Jihad Fars province).

Table 2. Descriptive statistics of the data for the soil fertility (Organization of Agriculture, Jihad Fars province)

| Statistic parameters | K | P (mg/kg) | Cu | Fe (mg/kg) | Mn (mg/kg) | OC | Zn |
|---|---|---|---|---|---|---|---|




| | (mg/kg) | | (mg/kg) | | | (mg/kg) | (mg/kg) |
|---|---|---|---|---|---|---|---|
| maximum | 666.00 | 30.00 | 2.00 | 15.00 | 52.50 | 1.65 | 3.00 |
| minimum | 137.00 | 2.00 | 0.20 | 1.00 | 2.80 | 0.18 | 0.10 |
| average | 313.73 | 13.94 | 0.97 | 4.54 | 14.77 | 1.01 | 0.65 |
| STDEV | 104.28 | 6.49 | 0.36 | 2.84 | 10.71 | 0.35 | 0.50 |

## 3. Materials and methods

In order to prepare the soil fertility maps by applying the OWA method, 45 sample soils were used. At first, the interpolation maps were created for each parameter by using the Inverse Distance Weighted (IDW) and then the fuzzy parameter maps were created for each parameter in order to make different risk levels by taking the best advantages of OWA. Each method description is as follows:

*3.1.* **Inverse Distance Weighted (IDW)**

The IDW model was used for interpolating the effective data in determining the soil fertility such as potassium (K), phosphor (P), copper (Cu), iron (Fe), manganese (Mn), organic carbon (OC) and zinc (Zn). The IDW interpolation explicitly implements the assumption that things which are close to each other and

more alike than those that are farther apart. To predict a value for any unmeasured location, IDW will use the measured values surrounding the prediction location. The assumed value of an attribute $z$ at any un-sampled point is a distance-weighted average of the sampled points lying within a defined neighborhood around that un-sampled point. Essentially, this value is a weighted moving

average (Burrough, et al., 1998):

$$\hat{z}(x_0) = \frac{\sum_{i=1}^{n} z(x_i) \, \mathrm{d}_{ij}^{-r}}{\sum_{i=1}^{n} \mathrm{d}_{ij}^{-r}} \qquad (1)$$



Where $x_0$ is the estimation point and $x_i$ are the data points within a chosen

neighborhood. Weights ($r$) are related to the distance by $d_{ij}$.

## 3.2. Ordered Weight Average (OWA)

OWA is a multi-criteria evaluation procedure. The nature of the OWA procedure
depends on some parameters which can be specified by fuzzy quantifiers. The GIS-

based multi-criteria evaluation procedures involve a set of spatially defined
alternatives and a set of evaluation criteria represented as map layers. According to
the input data (the criterion weights and the criterion map layers), the OWA
combination operator associates with the i- th location (e.g., raster or point) of a set of
order weights v = $v_1$, $v_2$, . . . , $v_n$ such that $v_j \in [0, 1]$, j=1,2,..,n, $\sum_{j=1}^{N} v_j = 1$, and it is defined

as follows (see Yager, 1988; Malczewski et al., 2003):

$$OWA_t = \sum_{j=1}^{N} \left( \frac{u_j v_j}{\sum_{j=1}^{n} u_j v_j} \right) z_{tf} \tag{2}$$

Where $z_{i1} \geq z_{i2} \geq . . . \geq z_{in}$ are the sequences obtained by reordering the attribute
values $a_{i1}$, $a_{i2}$, . . ., $a_{in}$, and $u_j$ is the criterion weight reordered based on  the
attribute value, $z_{ij}$. It is important to focus on the difference between the two types

of weights (the criterion weights and the order weights). The criterion weights are
assigned to the evaluation criteria in order to indicate their relative importance. All
locations on the *j-th* criterion map are assigned to the same weight of $w_j$. The order
weights are associated with the criterion values on the location-by-location basis.
They are assigned to the *i-th* location's attribute value in the decreasing order

without considering the origin of the criterion map . With different sets of order
weights, one can generate a wide range of OWA operators including the most often
used GIS- based map combination procedures: the weighted linear combination



(WLC) and Boolean overlay operations, such as the intersection (AND) and union (OR) (Yager, 1988; Malczewski et al., 2003). The AND and OR operators represent the extreme cases of OWA and they correspond to the MIN and MAX operators, respectively. The order weights associated with the MIN operator are: $v_n = 1$, and $v_j = 0$ for all other weights. Given the order weights, $OWA_i (MIN) = MIN_j (a_{i1}, a_{i2}, \ldots, a_{in})$. The following weights are associated with the MAX operator: $v_1 = 1$, and $v_j = 0$ for all other weights, and consequently $OWA_i (MAX) = MAX_j (a_{i1}, a_{i2}, \ldots a_{in})$. Assigning equal order weights (that is, $v_j = 1/n$ for $j = 1, 2, \ldots, n$) results in obtaining the conventional WLC which is situated at the mid-point on the continuum ranging from the MIN to MAX operators (Table 1) (Malczewski, 2006).

Table 1. Properties of the Regular Increasing Monotone (RIM) quantifiers with selected values of the Parameter (source: Malczewski, 2006).

| $\alpha$ | Quantifier ($Q$) | Order Weights($v$ik) | GIS Combination Procedure | ORness | rade-off |
|---|---|---|---|---|---|
| $\alpha \rightarrow =$ | At least one | Vi1=1; vik=0, (1<k<=n) | OWA (OR) | 1.0 | 0 |
| $\alpha$=0.1 | At least a few | a | OWA | a | a |
| $\alpha$=0.5 | A few | a | OWA | a | a |
| $\alpha$=1 | Half (identity) | vik=1/n , 1<=k<=n | OWA (WLC) | 0.5 | 1 |
| $\alpha$=2 | Most | a | OWA | a | a |
| $\alpha$=10 | Almost all | a | OWA | a | a |
| $\alpha \rightarrow \infty$ | All | Vin=1; vik=0, (1<=k<n) | OWA (AND) | 0 | 0.0 |

[a] The set of order weights depends on values of sorted criterion weights and parameter.

## 4. Results

### 4.1. Inverse Distance Weighted (IDW)

In the study area to determine the soil fertility, 45 sample points were used. These data were prepared by the Organization of Agriculture, Jihad Fars province in





2012. These points were collected by using a random sampling method of merely wheat fields. Because of the legal authority of some agriculture land owners in some parts of the study area, the points are not scattered well. In the study spline, the inverse distance weighted (IDW) and the simple Kriging method (Gaussian, circular, spherical, exponential model) were used to prepare raster maps for each soil parameter in ArcGIS 10.2. The results of the root-mean-square deviation (RMSE) for three models showed that the IDW method (circular model) with the lowest RMSE is the best model of the soil parameters prediction. According to Figure 2, sample points were selected randomly in the study area.

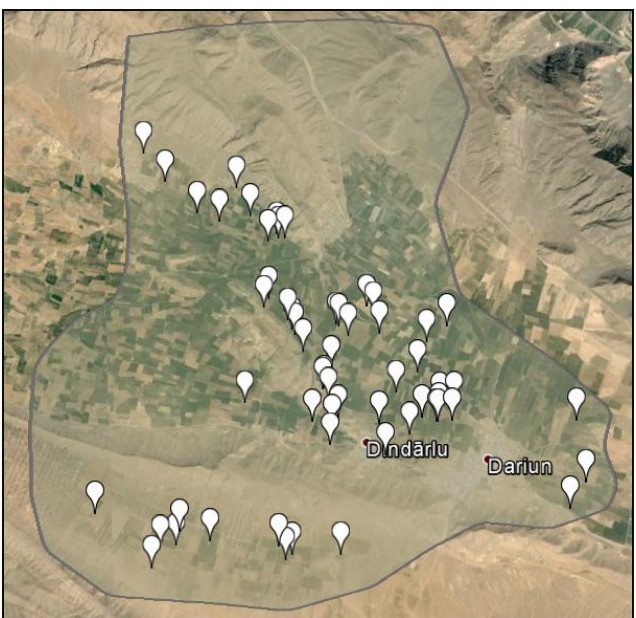

Figure 2. Position of sample points for the study area

In the study area, the IDW interpolation was used to predict K, P, Cu, Fe, Mn, OC and Zn values which are all shown in Figure 3. According to Figure 3, within the chosen study area, most elements in the north and parts of the south were determined to have lower amounts than other regions.

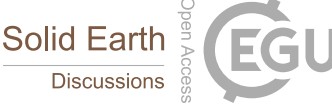

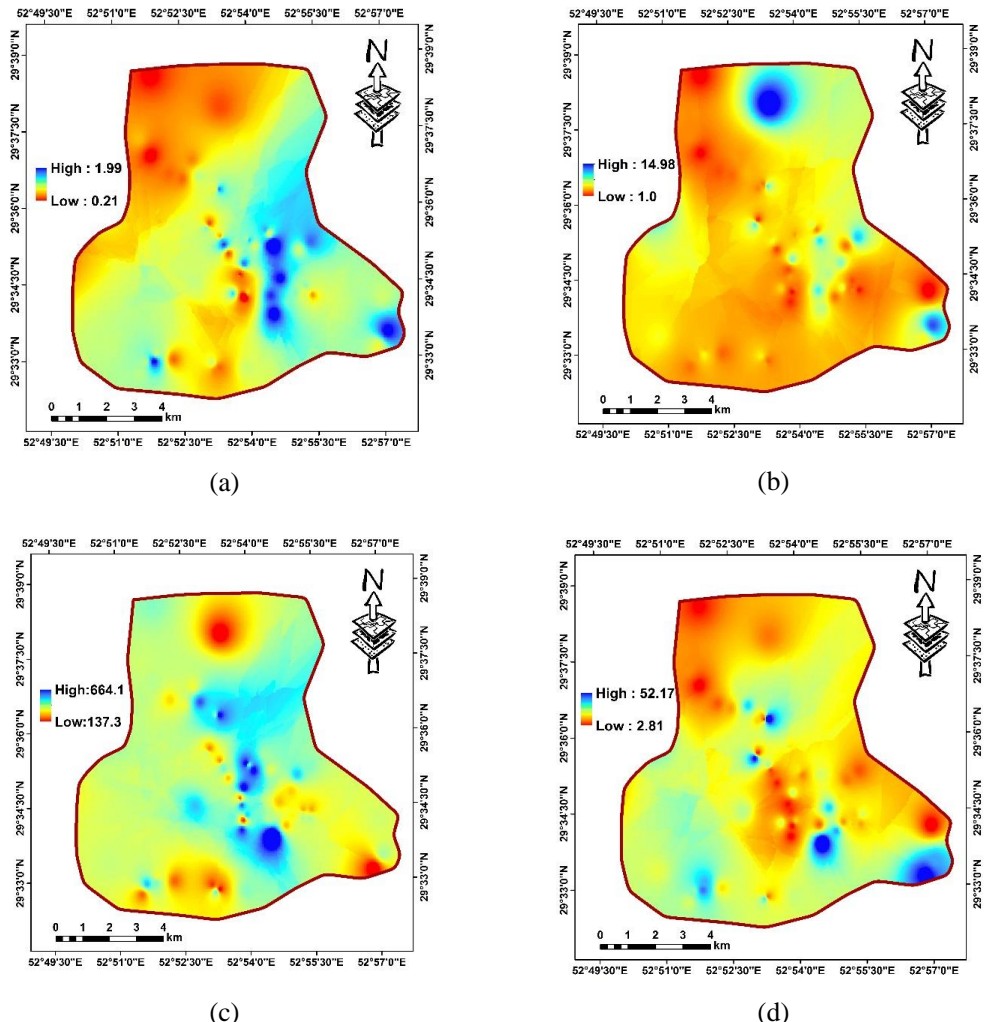





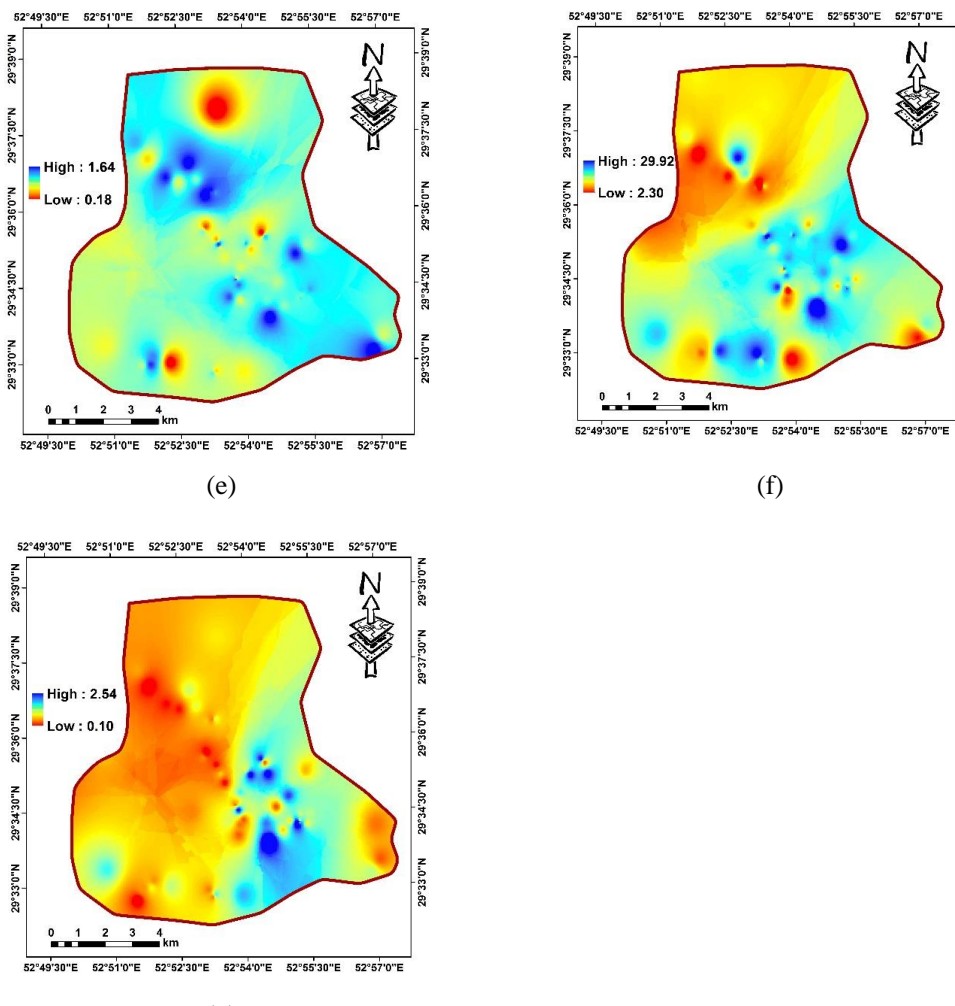

(g)

Figure 3. The interpolation map prepared by using IDW method. (a):K; (b):P (c):CU, (d):Fe; (e):Mn; (f):OC; (g): Zn.

### 4.2. Fuzzy method

In this study, K, P, Cu, Fe, Mn, OC and Zn maps from the IDW model were used as the input of the fuzzy inference system. In order to homogenize each parameter at first, maps are weighted by the OWA method and then the fuzzy method was




used for preparing the final soil fertility maps. According to FAO (1983), the membership function for each parameter was defined (K, P, Cu, Fe, Mn, OC and

Zn) and each fuzzy map was created for those elements which are between 0 and 1. The prepared fuzzy maps for the soil fertility parameters are shown in Figure 4. It must be noted that, by decreasing the soil fertility, MF is closer to 0 whereas by increasing the soil fertility, MF is closer to 1 (Soroush et al., 2011).

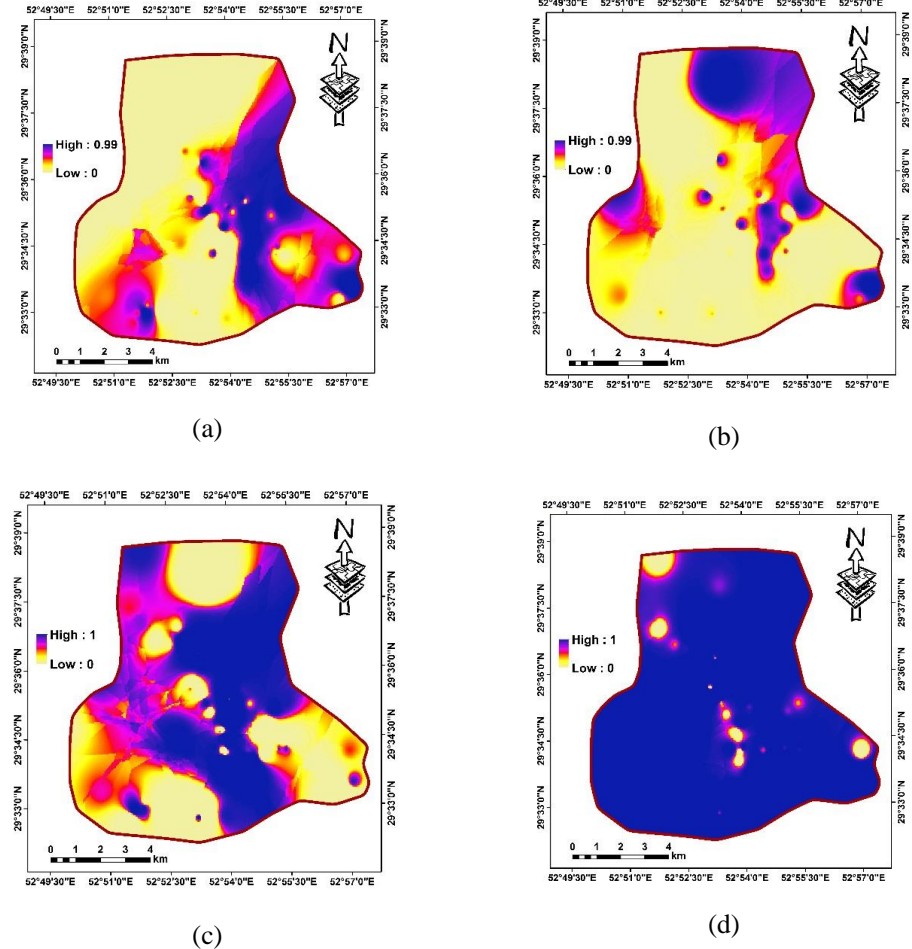

(a)                                    (b)

(c)                                    (d)





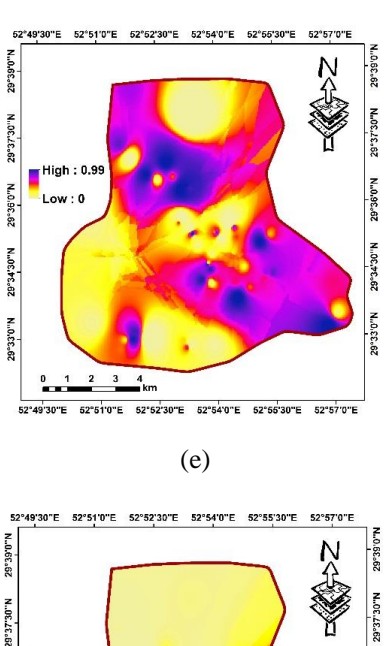
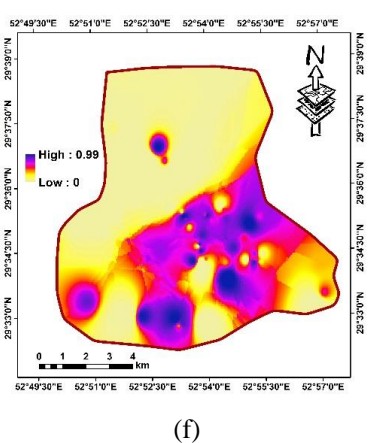

(e)                                         (f)

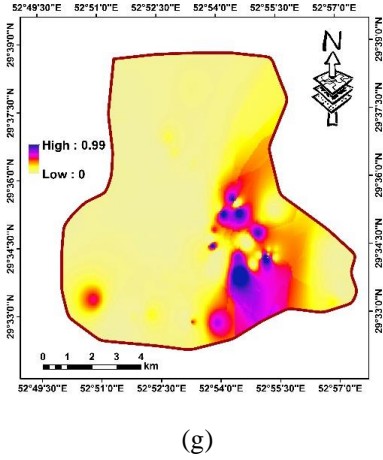

(g)

Figure 4. The fuzzy map of the studied area for each soil fertility parameter (a):K;
(b):P (c):CU, (d):Fe; (e):Mn; (f):OC; (g): Zn.

According to Figure 4, most of the study area did not have a suitable value for Mn
parameter which had a value close to zero (critical limit =10 (mg/kg)) in the fuzzy
map. However, results of the fuzzy method showed that most of the study area
(parts of the east, southeast and the small parts of the south west of the study area)
had suitable values for P and Zn parameters which had the value close to 1 in the
fuzzy map. Parts of the north, south west and south of the study area were not
suitable concerning the notion of fertility. According to the K fuzzy map, some



parts of the north, southeast and west were not suitable as well. Besides, some parts of the north, northwest and south of the study area were not suitable for
planting Cu. Finally, it was determined that only some parts of the northeast, southeast and small parts of the west and east were suitable concerning the soil fertility.

Finally to overt each parameter and to prepare the soil fertility, the OWA method
was used. OWA offers a wealth of possible solutions for our residential developmental problems. In our application, six order weights were applied corresponding to seven factors which were rank-ordered for each parameter after utilizing the modified factor weights. Table 3 gives six typical sets of order weights for seven factors: (1) an average level of the risk and a full trade-off, (2) a
low level of the risk and no trade-off, (3) a high level of the risk and no trade- off, (4) a low level of the risk and an average trade-off, (5) a high level of the risk and an average trade-off, (6) an average level of the risk and no trade-off. Figure 5 shows the locations of typical sets of order weights in the decision-support space (Figure 5).

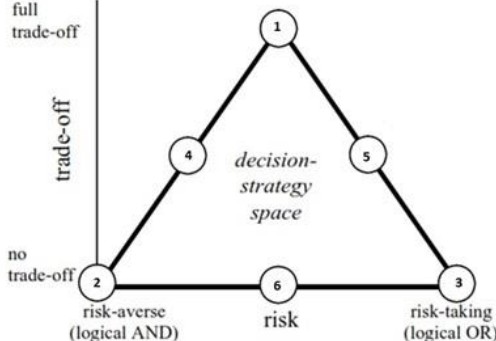


Figure 5. The decision-strategy space and typical sets of order weights (see Table 3)





Table 3: Typical sets of order weights for seven factors*.

| Rank | 1st order weight | 2nd order weight | 3rd order weight | 4th order weight | 5th order weight | 6th order weight | 7th order weight |
|---|---|---|---|---|---|---|---|
| 1 Average level of risk and full trade-off | 0.1428 | 0.1428 | 0.1428 | 0.1428 | 0.1428 | 0.1428 | 0.1428 |
| 2 Low level of risk and no trade-off | 1 | 0 | 0 | 0 | 0 | 0 | 0 |
| 3 High level of risk and no trade-off | 0 | 0 | 0 | 0 | 0 | 0 | 1 |
| 4 Low level of risk and average trade-off | 0.4455 | 0.2772 | 0.1579 | 0.0789 | 0.0320 | 0.0085 | 0 |
| 5 High level of risk and average trade-off | 0 | 0.0085 | 0.032 | 0.0789 | 0.1579 | 0.2772 | 0.4455 |
| 6 Average level of risk and no trade-off | 0 | 0 | 0 | 1 | 0 | 0 | 0 |

* The order of elements is as following:1st:OC, 2nd:P, 3rd:K, 4th:Zn, 5th:Fe,6th:Cu,7th:Mn


Concerning the standardized criterion maps and the corresponding criterion weights, we apply the OWA operator using Eq. (2) for selected values of fuzzy quantifiers. These values are attributed as: at least one, at least a few, a few, identity, most, almost all, and all. Each quantifier is associated with a set of order

weights which are calculated according to Eq. (2). Figure 6 shows six alternative soil fertility patterns.

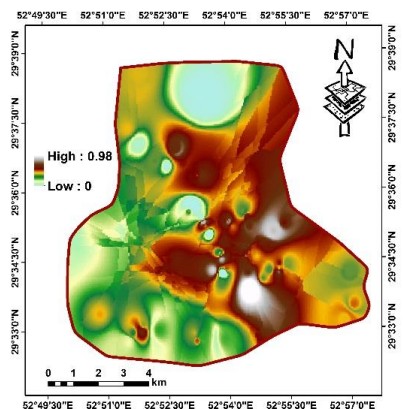

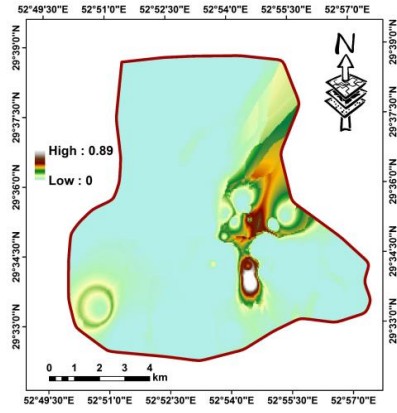

(2)



(1)

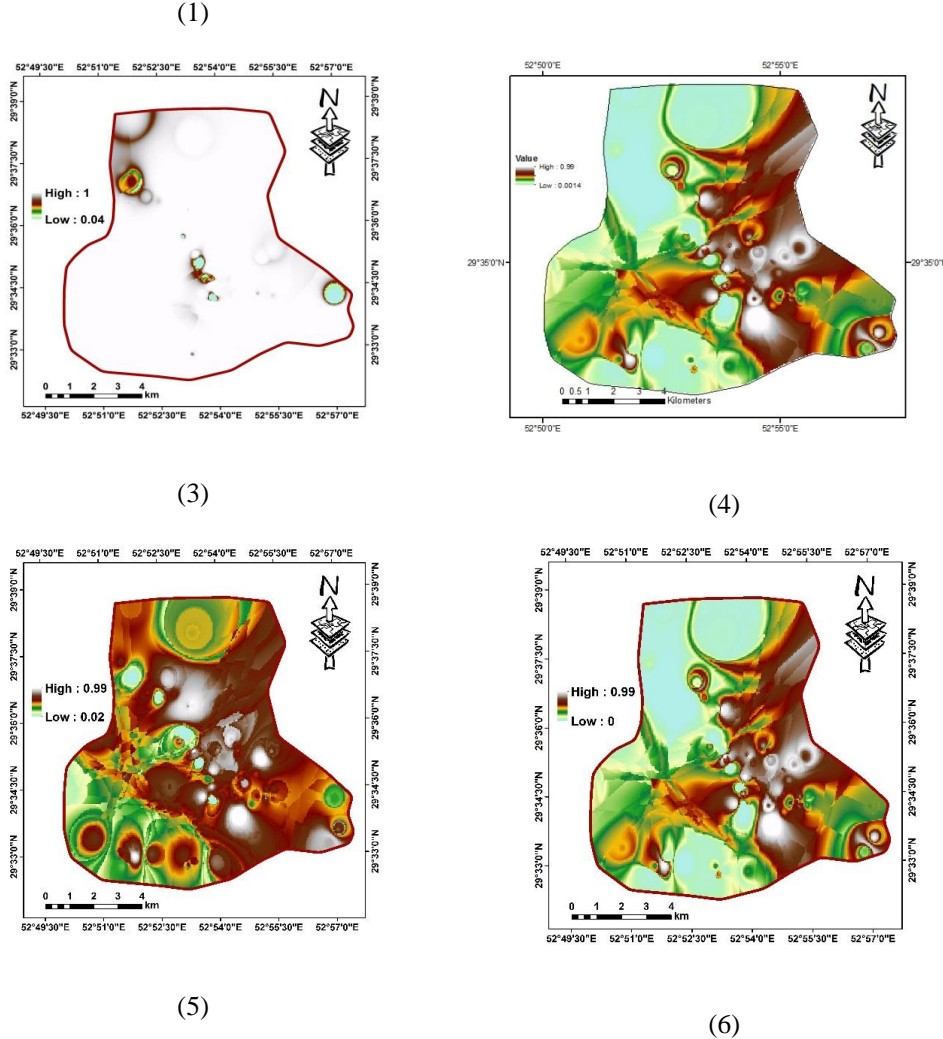

(3)                                                    (4)

(5)                                                    (6)

Figure 6. The soil fertility maps of OWA results for the selected fuzzy linguistic quantifiers. (1): An average level of the risk and a full trade-off, (2): A low level of the risk and no trade-off, (3): A high level of the risk and no trade-off, (4): A low level of the risk and an average trade-off, (5): A high level of the risk and an average trade-off, (6): An average level of the risk and no trade-off

According to Figure 6 (1), parts of the study area had a high value for the soil fertility (a high risk level for farmers who have good financial conditions). According to Figure 6 (2), with decreasing the risk (no trade-off) the area with the

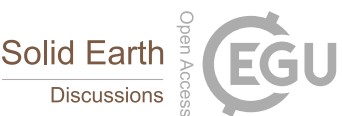

high soil fertility could be determined. Therefore, only parts of the east, northeast and southwest of the study area were suitable for the soil fertility. Nevertheless, almost all parts were not suitable for the soil fertility. According to Figure 6 (3), almost the whole study area had the low soil fertility. Figure 6 (4) showed a low

risk with an average trade-off that had more risk in comparison to Figure 6 (2). Figure 6 (5) showed a high risk with an average trade-off that had a lower risk in comparison to Figure 6 (3) done for determining the soil fertility. Figure 6 (6) showed an average risk with no trade-off that had more risk in comparison to Figure 6 (3).

**5. Discussion**
Based on Table 4, the OWA map was classified into eight classes that are shown in Figure 7, figure 8 and Table 5. Figure 7 shows six alternative soil fertility patterns. According to Figure 7 with an average risk (a full trade-off) (Figure 7 (1)), all effective parameters of the soil fertility received the same weight (0.33).

According to Figure 7 (1), parts of the study area either had a high value (the southeast and the east of the study area), or a low value (the north of the study area). According to Figure 7 (2), with decreasing the risk (no trade-off), the area with the high soil fertility can be determined. Therefore, just the east parts of the study area were suitable for the soil fertility whereas almost all parts were not

suitable for the soil fertility. Moreover by increasing the risk (no trade-off) (Figure 7 (3)), almost all parts of the study area had a good soil fertility. Figure 7 (4) showed a low risk with an average trade-off that had more risk in comparison to Figure 7 (2). Figure 7 (5) showed a high risk with an average trade-off that showed a lower risk for determining the soil fertility in comparison to Figure 7

(3). Figure 7 (6) showed an average risk with no trade-off that demonstrated more risk (0-1) in comparison to Figure 7 (3).



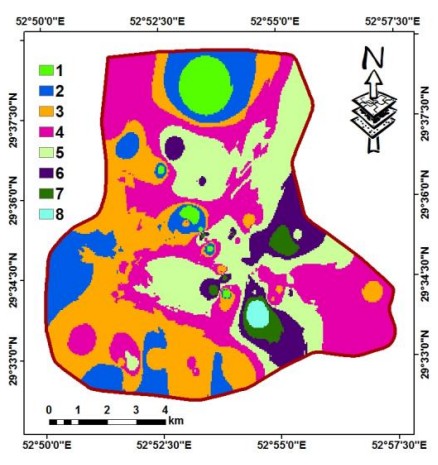

(1)

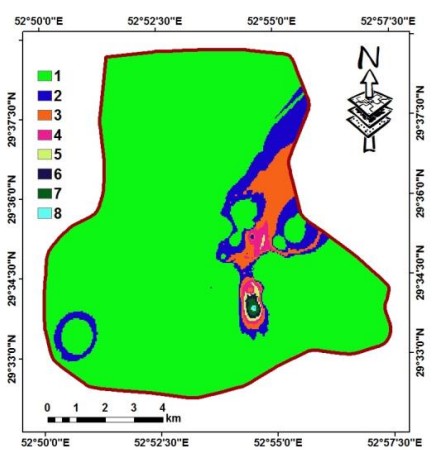

(2)

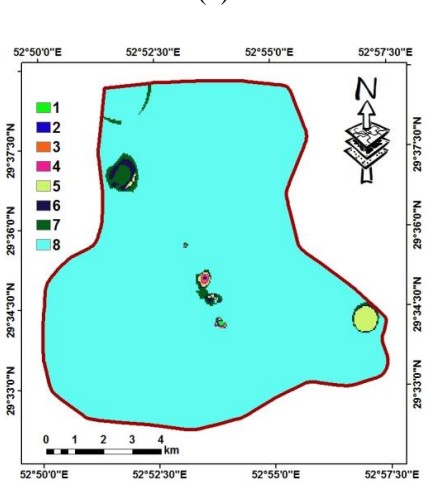

(3)

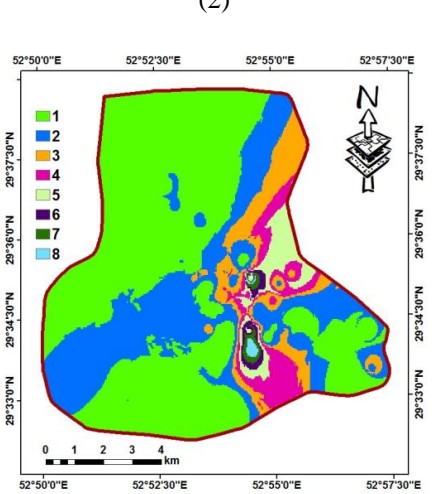

(4)





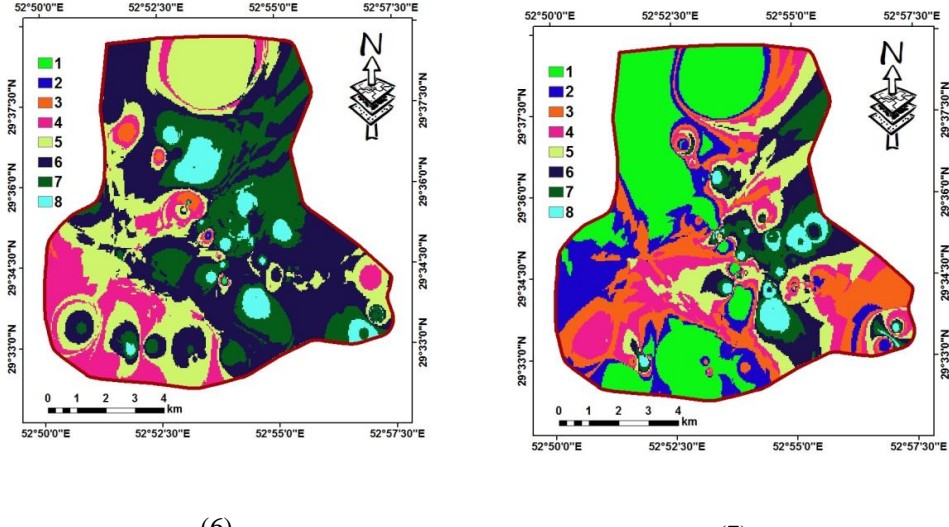

(6)                                    (7)

**Figure 7.** The OWA map was classified into eight classes. (1): An average level of the risk and a full trade-off, (2): A low level of the risk and no trade-off, (3): A high level of the risk and no trade-off, (4): A low level of the risk and an average trade-off, (5): A high level of the risk and an average trade-off, (6): An average level of the risk and no trade-off

Table 4. Description of each classes for soil fertility

|   | Range | Description |
|---|-------|-------------|
| 1 | 0 – 0.125 | Extremely low |
| 2 | 0.125 – 0.25 | Very low |
| 3 | 0.25 – 0.375 | Low |
| 4 | 0.375 – 0.5 | Moderately Medium |
| 5 | 0.5 – 0.625 | Medium |
| 6 | 0.625 – 0.75 | High |
| 7 | 0.75 – 0.875 | Very high |
| 8 | 0.875 - 1 | Extremely high |






Based on Table 5, the OWA map was classified into eight classes. Results of the present study are similar to the results of the other research done by Mokarram and Aminzadeh (2010). They used seven order weights for determining the land suitability. Besides, they applied ten corresponding factors (EC, pH, ESP, CaCO₃,

Gypsium, wetness, texture, slope, depth and topography) which were rank-ordered for each parameter. Drobne and Lisec (2009) used OWA for determining seven factors of the soil fertility analysis and obtaining six designs with different risk level. In fact, using OWA can produce an almost infinite range of possibilities used for different designs. Newest researches in the field of

agricultural issues such as the soil fertility are done by Khaki et al. (2015), Bijanzadeh and Mokarram (2013) and Mokarram, Bardideh (2012) which can determine the soil fertility by applying the fuzzy algorithm. In this research, only the medium risk (AHP) was used and researchers did not check different risk levels. Totally, it is stated that using the OWA method with difference risk levels

can create several maps that can help a user (for example farmer) to make different decisions according to different financial situations and different risk levels. For example with a low risk, the farmer can select an area that has more soil fertility to yield maximum products. Thus, OWA can be applied in the fields of the natural science to provide the suitable condition which helps making

accurate decisions.

Table 5. Area (km²) of each class by using the OWA method for the soil fertility

| class | (1) | (2) | (3) | (4) | (5) | (6) |
|---|---|---|---|---|---|---|
| 1 | 3.39 | 87.98 | 0.03 | 56.48 | 0.05 | 27.70 |
| 2 | 12.68 | 6.97 | 0.04 | 25.82 | 0.10 | 13.89 |
| 3 | 24.45 | 3.62 | 0.04 | 7.82 | 0.84 | 15.69 |
| 4 | 29.23 | 0.87 | 0.06 | 5.09 | 12.55 | 13.20 |




| 5 | 21.26 | 0.25 | 0.69 | 3.41 | 25.17 | 11.20 |
| 6 | 6.60 | 0.16 | 0.59 | 0.73 | 36.69 | 9.38 |
| 7 | 1.83 | 0.15 | 1.34 | 0.44 | 19.76 | 6.25 |
| 8 | 0.59 | 0.03 | 97.23 | 0.24 | 4.86 | 2.70 |

Yu et al. (2010) used GIS-based AHP-SA tool and MCDM model for the irrigated cropland suitability assessment addressing. It demonstrated that the tool was spatial, simple and flexible.

Moreover, Yu et al. (2011) used a new CA-based spatial multi-criteria evaluation (MCE) methodology to conduct the land suitability simulation (LSS). This, in turn, could help the decision-makers to optimize the land allocation and to make better land-use planning decisions.

Chen et al. (2013) developed a unique methodology which extends the AHP-SA model proposed to increase the efficiency, improve the flexibility and enhance the visualization capability; besides, the spatial framework was developed as AHP-SA2 within a GIS platform. It assisted stakeholders and it researched into arriving at a better understanding of the weight sensitivity for characterizing, reporting and minimizing the uncertainty in the AHP-based spatial MCDM.

## 6. Conclusions

The soil fertility problem involves the evaluation and classification of the areal units according to their fertility for a particular activity. Therefore, the aim of the present study was to produce the soil fertility maps based on OWA operators of the GIS-based multi-criteria evaluation procedures in the southeast of Iran. The OWA approach provides a mechanism for guiding the decision maker/analyst through the multi-criteria combination procedures. The OWA method is an important tool in the management sciences and operational researches. Types of decision rules with



definitions in OWA method lead to solve semi-structured decision problems. In order to prepare the soil fertility by using OWA at first, the IDW model was determined to interpolate maps into the input data such as potassium (K), phosphor (P), copper (Cu), iron (Fe), manganese (Mn), organic carbon (OC) and zinc (Zn). Then, K, P, Cu, Fe, Mn, OC and Zn maps from IDW were used as the input of the

fuzzy inference system. Finally, in order to prepare the standardized criterion maps and corresponding criterion weights, the OWA operator was applied to select values of fuzzy quantifiers known as : least one, at least a few, a few, identity, most, almost all, and all . Each quantifier is associated with a set of order weights which are calculated as well. Results showed that with decreasing the risk (no

trade-off), the area with a high soil fertility was determined. Therefore, just parts of the east and southeast of the study area were considered suitable for the soil fertility. Furthermore, with increasing the risk (no trade-off), almost all of the study area had a good soil fertility. Thus, the OWA method is able to prepare high maps of the soil fertility with different managements.

**Acknowledgements**

The authors would like to acknowledge the Organization of Agriculture, Jihad Fars for their assistance during conducting the present study by providing the required dataset.

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



**Figure captions**

Figure 1. Location of the study area (digital elevation model (DEM) with spatial resolution of 30 m) (Source: http://earthexplorer.usgs.gov).

Figure 2. Position of sample points for the study area.

Figure 3. Interpolation map using IDW method. (a):Cu; (b):Fe; (c):K; (d):Mn; (e):OC; (f):P; (g): Zn.

Figure 4. Fuzzy map of studied area for each soil fertility parameter. (a):Cu; (b):Fe; (c):K; (d):Mn; (e):OC; (f):P; (g): Zn.

Figure 5. Decision-strategy space and typical sets of order weights (see Table 3)

Figure 6. Soil fertility maps of OWA results for selected fuzzy linguistic quantifiers

Figure 7. Classification of OWA map for soil fertility.