# Peer review of "USING ORDERED WEIGHT AVERAGING (OWA) FOR MULTICRITERIA SOIL FERTILITY EVALUATION BY GIS (CASE STUDY: SOUTHEAST IRAN)"

_Solid Earth, 2016_

## Referee Comment (RC1) · Anonymous Referee #1 · 10 Sep 2016

English use in the manuscript is quite poor, it needs a careful editing by an editing service or a native English speaker.

The references are weak. Many (probably most, but I didn't do exact counts) are from non-Thompson Reuters indexed sources. This is not typical best practice when trying to publish in an indexed journal. To be acceptable for publication in a leading journal, the references will need to be significantly strengthened. In addition, there are some journal names I am not familiar with and when I tried to look them up I could not find them. That means the journals are either 1) very obscure or 2) the authors have not been careful in entering their references. Neither case is encouraging in a manuscript submitted for peer review in a strong journal.

[Figure]

It is not possible to determine the data quality with the information given, and if the data is not quality data, the maps generated are worthless. More information is needed about sampling depth, size of the sampling area (ha or square km) and testing methods used to determine soil fertility parameters. I know the authors did not run the fertility analyses, but there must be information about this in the government report.

The Materials and Methods, Results, and Discussion sections are intermixed. Each has material in it that belongs in one of the other sections.

Detailed comments: Page 4 – What is a "moderate climate"? And why is the climate first described as moderate, but then described as hot and semi-arid just a couple lines later. "moderate" and "hot" are relative terms and not very scientific. Give average temperature and precipitation values, and/or use climate categories from an established system such as the Köppen climate classification.

Page 4 – What is the relevance of the Shiraz population or the statement that the majority of the population is Persian? I did not see anywhere in the manuscript where this information was important. I suggest deleting it.

Page 5 – Why are these paragraphs numbered? Numbers need to be removed.

Figure 1 – There needs to be a map of Iran that shows where in the country the study area is located. This can be an inset as part of an expanded figure 1. Solid Earth is an international journal, and as currently presented most readers would have no idea where in Iran the study site is.

Page 7 – For each soil fertility parameter, report how it was determined. There are many different ways to determine organic carbon and P, for example. Also, are these total values, exchangeable values, etc.? Information like this is very important to understanding this study.

Table 1 belongs in the Results section. Even though you did not determine these soil fertility values, they were determined using techniques you need to report, and they

are results that you utilized in this study.

Page 10 – In line 198 it says "the points are not scattered well". Then in lines 204 it says "sample points were selected randomly". These two statements are a bit confusing, especially because of the order in which they are presented. In addition, in line 196-197 it says "using a random sampling method of merely wheat fields". If only wheat fields were sampled, this was not a truly random sampling design. This section needs to be rewritten, clarified, and conflicting statements (random sampling versus sampling only of wheat fields) need to be rectified.

Figure 3 – What are the units for each of the soil fertility measurements? Also, the latitude/longitude measurements along the sides of the figures are too small to read. This second statement is also tru of Figure 4.

Page 14 – At the bottom of the page it talks about the fuzzy method maps. In this discussion, it says that Mn values were close to 0, while P and Zn parameters were close to 1. However, map g is Zn, and as I look at map g, most of the values are close to 0, not close to 1. On the other hand, map e is Mn, and much more of map e is close to 1 than of map g. The discussion of these maps needs to be carefully reconsidered and rewritten, because right now the maps do not correspond to the discussion.

Page 15 – Conclusions are made here regarding overall soil fertility status, but soil fertility status is based on an assemblage of soil fertility parameters, not on single parameters. This study would be much more robust if the authors would use GIS to overlay the fuzzy maps and generate new maps that show locations that are high in all (or most) fertility parameters versus areas low in all (or most) fertility parameters.

Page 15, Lines 243-247 – How were the weights determined for the seven fertility factors? This should be discussed in detail in the Materials and Methods. As currently given, you have simply assigned weights with no real explanation and asked the reader to accept them. In such a situation, the reader cannot independently evaluate the validity of your assigned weights.

Page 16, Lines 255-261 – This section also needs much more explanation in the Materials and Methods section.

Pages 17-18, Lines 268-279 – I don't follow this. It needs better explanation.

Page 18 – The first paragraph of the Discussion belongs in Materials and Methods.

The overall discussion is weak. It does not show how this study adds to our body of knowledge and compare and contrast the findings from this study to the findings of other similar studies.

---

## Referee Comment (RC2) · Anonymous Referee #2 · 22 Sep 2016

In this paper, the concerns of articles, which are about how to incorporate the Inverse Distance Weighted (IDW) and Ordered Weighted Averaging(OWA) to make the multicriteria soil fertility maps based on the GIS. It aims to introduce a method for farmers in case of make balance between their budget and their farm soil parameters.

Weakness: 1. The layout of samples is unreasonable, in the northeast and west part of which has no any sample. Moreover, the author did not verify the accuracy of the interpolation results in that part. Accordingly, it is doubtful in terms of the accuracy of their final results and the possibility of the agricultural application.

2. Although the method is good, it is only a general application of the methods, which has no much new ideas in the paper. Furthermore, the paper failed to explain how to

make decision for the farmers based on these different criteria findings, which reduces the value of this paper.

3. One of the main weaknesses is its language. It is really hard to understand.

In details: 1. Line 127. Please provide additional information of the speciation of mineral elements in this paper. Does it means the total amount, available amount or others? 2. Tab 2. Please provide additional information on the chemical methods used in measuring these factors. 3. Tab 2. Content of OC is out of normal range. Please check the value and the unit. 4. Fig 2. Please add map elements in the Fig. 5. Fig 3. Please add the unit for the factors. 6. Line 268-269. I can hardly understand its meaning.

In summary, in this article, the idea is not innovative, the meaning is general, the design is defective, and the reliability of the results is in doubt. Thus, I would recommend to reject it.

---

## Referee Comment (RC3) · Anonymous Referee #3 · 26 Sep 2016

General comments:

In this manuscript, Marzieh Mokarram and Majid Hojati present a study on the analysis of soil fertility by multi-criteria analysis, i.e. Ordered Weighted Averaging in the west of Shiraz, Fars province in Iran. The manuscript is in general not well organised and the contents of the different sections are intermixed. The methods are not adequately explained and lack accuracy. The discussion needs some work, e.g. the authors should discuss their results in a more global context using more updated references and argue about the application of these results at different scales by decision-makers. Also, the manuscript needs to be revised by an English native speaker as it contains several grammar errors. See below some specific comments.

[Figure]

Specific comments:

Abstract: - Line 12: remove 'The' and 'the'. - Line 14: what do you mean with 'effective factors of the natural science'? - Line 23: the sentence 'a farmer can accept more risk it can use more areas...' does not make a lot of sense. - Lines 29-30. Rewrite that sentence, does not make sense ('..by decreasing the risk (no trade-off), increasing the risk..' Keywords: I suggest using different words than those from the title.

Introduction:

More updated references that not only focus on the area of Iran should be included here. Page 2. Lines 41-42. The following references should be in chronological and then alphabetical order. Please check across the manuscript, e.g. Salehi et al., 2012; Feng et al., 2012; Ashrafi et al., 2012. Page 3, lines 56-58. The objectives of the study should go at the end of this section.

Study area.

This section should be included within the 'Materials and Methods' section. Please check the manuscript preparation guidelines of the journal: http://www.solid-earth.net/for_authors/manuscript_preparation.html Please, be specific with the climatic classification of the study area. 'This city climate is categorized as a hot semi-arid one' is not very clear. And the same goes for 'Winters are cool with an average of freezing temperature in December and January'. What are the mean/average temperatures and the annual rainfall?

Page 4; line 96-97 The majority of Shiraz populations are Persian. This is not really relevant to the study, is it?

Pages 4-5; lines 102-120. Why do you use numbering here? It is confusing.

Figure 1. This figure does not reveal much. Where is the study area located within the country?

Methods/Results sections:

Table 2. How were these parameters measured? Number of replicates, methods, equipment, etc.. Methods rely on these parameters but you do not provide any information... Moreover, the actual explanation of your methods (e.g. soil sampling, modelling, etc.) is included in the results section instead. Please, organise your sections according to the guidelines. In any case, the methods are not fully reliable, as the authors do not test the interpolation and the baseline information is not explained in detail and its quality cannot be determined. Also the sampling method is confusing. Was it a random sampling or a sampling focalised on wheat fields? The weighting method is also not explained. Please specify all these issues.

Discussion

The discussion is very poor. You need to elaborate on this section and connect your results (maps) to properly discuss the fertility status of these soils. How is this study important to decision-makers?

---

## Author Comment (AC1) · 7 Nov 2016

Dear Editor Allow us to thank the reviewers for their comments, suggestions and corrections that were helped to revise to manuscript. The manuscript was revised for use of English language and syntax while the innovation was outlined in our responses to reviewer's comments. the last changes and response to comments are attached int this message Best wishes

Please also note the supplement to this comment:
http://www.solid-earth-discuss.net/se-2016-112/se-2016-112-AC1-supplement.zip